# Using machine learning to understand age and gender classification based on infant temperament

**Maria A. Gartstein**[1]*, **D. Erich Seamon**[2], **Jennifer A. Mattera**[1], **Michelle Bosquet Enlow**[3], **Rosalind J. Wright**[4,5], **Koraly Perez-Edgar**[6], **Kristin A. Buss**[6], **Vanessa LoBue**[7], **Martha Ann Bell**[8], **Sherryl H. Goodman**[9], **Susan Spieker**[10], **David J. Bridgett**[11], **Amy L. Salisbury**[12], **Megan R. Gunnar**[13], **Shanna B. Mliner**[13], **Maria Muzik**[14], **Cynthia A. Stifter**[6], **Elizabeth M. Planalp**[15], **Samuel A. Mehr**[16], **Elizabeth S. Spelke**[16], **Angela F. Lukowski**[17], **Ashley M. Groh**[18], **Diane M. Lickenbrock**[19], **Rebecca Santelli**[20], **Tina Du Rocher Schudlich**[21], **Stephanie Anzman-Frasca**[22], **Catherine Thrasher**[23], **Anjolii Diaz**[24], **Carolyn Dayton**[25], **Kameron J. Moding**[26], **Evan M. Jordan**[27]

1 Washington State University, Pullman, WA, United States of America, 2 University of Idaho, Moscow, ID, United States of America, 3 Boston Children's Hospital and Harvard Medical School, Boston, MA, United States of America, 4 Department of Pediatrics, Kravis Children's Hospital, New York, NY, United States of America, 5 Icahn School of Medicine at Mount Sinai, New York, NY, United States of America, 6 Pennsylvania State University, University Park, PA, United States of America, 7 Rutgers University, New Brunswick, NJ, United States of America, 8 Virginia Tech, Blacksburg, VA, United States of America, 9 Emory University, Atlanta, GA, United States of America, 10 University of Washington, Seattle, WA, United States of America, 11 Northern Illinois University, DeKalb, IL, United States of America, 12 Virginia Commonwealth University, Richmond, VA, United States of America, 13 University of Minnesota, Minneapolis, MN, United States of America, 14 University of Michigan, Ann Arbor, MI, United States of America, 15 University of Wisconsin, Madison, WI, United States of America, 16 Harvard University, Boston, MA, United States of America, 17 University of California, Irvine, CA, United States of America, 18 University of Missouri, Columbia, MO, United States of America, 19 Western Kentucky University, Bowling Green, KY, United States of America, 20 University of North Carolina, Chapel Hill, VA, United States of America, 21 Western Washington University, Bellingham, WA, United States of America, 22 University of Buffalo, Buffalo, NY, United States of America, 23 University of Virginia, Charlottesville, VA, United States of America, 24 Ball State University, Muncie, IN, United States of America, 25 Wayne State University, Detroit, MI, United States of America, 26 Purdue University, West Lafayette, IN, United States of America, 27 Oklahoma State University, Stillwater, OK, United States of America

* gartstma@wsu.edu

**Data Availability Statement:** Data cannot be shared publicly because of concerns regarding potentially identifying details (i.e., age and gender being linked to sensitive participant information

## Abstract

Age and gender differences are prominent in the temperament literature, with the former particularly salient in infancy and the latter noted as early as the first year of life. This study represents a meta-analysis utilizing Infant Behavior Questionnaire-Revised (IBQ-R) data collected across multiple laboratories ($N$ = 4438) to overcome limitations of smaller samples in elucidating links among temperament, age, and gender in early childhood. Algorithmic modeling techniques were leveraged to discern the extent to which the 14 IBQ-R subscale scores accurately classified participating children as boys ($n$ = 2,298) and girls ($n$ = 2,093), and into three age groups: youngest (< 24 weeks; $n$ = 1,102), mid-range (24 to 48 weeks; $n$ = 2,557), and oldest (> 48 weeks; $n$ = 779). Additionally, simultaneous classification into age and gender categories was performed, providing an opportunity to consider the extent to which gender differences in temperament are informed by infant age. Results indicated that overall age group classification was more accurate than child gender models, suggesting

(i.e., temperament scores). This determination was made by several Research Ethics Committees/Institutional Review Boards at the home institutions of researchers contributing data to the present meta-analytic effort. Data requests may be sent to Washington State University (home institution of the corresponding author, who compiled the dataset): Dan Nordquist, Associate Vice President of Research, Lighty Student Services Bldg, Room 280, PO Box 641060, Pullman, WA 99164-1060; email: orso@wsu.edu; phone: 509 335-9661.

**Funding:** MBE: R01HL095606, R01HD082078; National Institutes of Health https://www.nih.gov KPE, KB, & VL: NIH R01 MH109692; R21 MH103627 National Institutes of Health https://www.nih.gov MAB: R01 HD049878; R03 HD043057 National Institutes of Health https://www.nih.gov SG: 1P50 MH58922-01A1; 1P50 MH077928-01A1 National Institutes of Health https://www.nih.gov SS: 5R01HD080851-05 National Institutes of Health https://www.nih.gov AS: R01MH78033 National Institutes of Health https://www.nih.gov DL: 8P0GM103436; P20GM103436; 8P20GM103436 National Institutes of Health https://www.nih.gov TDRS: MFS 901; MFS 907 Western Washington University https://www.wwu.edu SAF: DK72996; M01RR10732 National Institutes of Health https://www.nih.gov None of the funders had any role in study design, data collection and analysis, decision to publish, or preparation of the manuscript.

**Competing interests:** The authors have declared that no competing interests exist.

that age-related changes are more salient than gender differences in early childhood with respect to temperament attributes. However, gender-based classification was superior in the oldest age group, suggesting temperament differences between boys and girls are accentuated with development. Fear emerged as the subscale contributing to accurate classifications most notably overall. This study leads infancy research and meta-analytic investigations more broadly in a new direction as a methodological demonstration, and also provides most optimal comparative data for the IBQ-R based on the largest and most representative dataset to date.

## Introduction

Although a number of approaches have been developed for the purpose of measuring temperament in childhood, including a variety of observational procedures and physiological techniques, parent report continues to be most widely used overall [1]. The latter is due to a number of factors, prominently among these being ease of administration and scoring as well as accessibility. Parent-report also provides descriptors of child temperament across time and situations, not just a "snapshot" of reactivity and/or regulation that can be gleaned from brief laboratory observations. Although multiple temperament theories or frameworks have been proposed, Rothbart's psychobiological model is generally viewed as most widely accepted at this time [2]. This approach casts temperament as constitutionally based individual differences in reactivity and self-regulation, with constitutional referring to the relatively enduring biological make-up of the individual, influenced by heredity, maturation, and experience. Reactivity refers to the arousability of emotional, motor, and attentional responses, assessed by threshold, latency, intensity, time to peak intensity, and recovery time of reactions. Self-regulation embodies processes that can serve to modulate reactivity, such as soothability and inhibitory control [3].

Although temperament has often been delineated into three overarching factors of Negative Emotionality, Positive Affectivity/Surgency, and Regulatory Capacity/Orienting, more recent studies emphasize the narrowly defined component scales. This shift toward a fine-grained approach is a function of research demonstrating individual scales that belong to the same overarching factor differentially predict important outcomes (e.g., behavior problems), present with growth trajectories discrepant from the overarching factors, and contribute to temperament profiles in a manner inconsistent with the overarching factor content (i.e., scales that load onto different factors contribute to the same profile, and vice versa–components of the same factor define different profiles/classes; [4–7]. The Infant Behavior Questionnaire-Revised (IBQ-R) designed to provide indicators of infant temperament comprises 14 fine-grained scales: Activity Level, Smiling/Laughter, Approach, High Intensity Pleasure, Perceptual Sensitivity, Vocal Reactivity, Fear, Distress to Limitations, Sadness, Falling Reactivity, Duration of Orienting, Soothability, Cuddliness/Affiliation, and Low Intensity Pleasure, and is the focus of this investigation.

### Development of temperament and age differences

Manifestations of temperament transform over development, with rapid change during infancy [8]. Positive emotionality (e.g., smiling), rarely expressed during the newborn period, is observed more reliably between ages two and three months, and increases in expression throughout the first year of life [8,9]. Levels of activity, approach, distress to limitations, and

fear increase throughout the first year of life as well [10–14]. Anger reactions across infancy appear to follow a U-shaped trajectory [12,15]. The decrease in anger responses occurring between 2 and 6 months of age has been linked to greater flexibility in attention shifting [16]. In the second half of the first year, infants are likely to respond with anger when unable to grasp an attractive stimulus that has been placed out of reach, or when a caregiver has removed a forbidden object. Fear generally increases throughout the second half of the first year of life [10,12–14], with inhibition of approach toward novel and/or intense stimuli "coming online" [14,17].

The developmental course of attentional orienting has been described as U-shaped in the first year of life [18]. Carranza and colleagues [12], for example, noted decreases in Duration of Orienting between 6 and 9 months, followed by an increase between 9 and 12 months. Toward the end of the first year, skills associated with the development of the executive attention system may contribute to the flexibility of orienting reactions [19–21]. Infants also gain communication skills rapidly during the first year of life [22,23], and thus exhibit greater vocal reactivity over time.

With respect to age/developmental differences discerned via the IBQ-R, older infants obtain higher scores on Approach, Vocal Reactivity, High Intensity Pleasure, Activity Level, Perceptual Sensitivity, Distress to Limitations, and Fear, whereas younger infants' scores are higher for Low Intensity Pleasure, Cuddliness/Affiliation, and Duration of Orienting [24,25]. More recent longitudinal investigations provided further evidence of increases in Fear across the first year of life [5,26], also noting increases in Distress to Limitations and Sadness, albeit not always linear in nature. Falling Reactivity was associated with a quadratic trajectory, with increases followed by declining values later in infancy. Increasing trajectories were noted for attributes associated with Positive Affectivity/Surgency, with trends toward greater Activity Level, Smiling and Laughter, High Intensity Pleasure, Approach, Perceptual Sensitivity, and Vocal Reactivity later in infancy. Growth modeling provided evidence of nonlinear changes in Duration of Orienting, Soothability, Cuddliness, and Low Intensity pleasure, wherein initial growth in values was followed by decreases later in infancy [5]. These findings are largely consistent with prior research relying on different measurement approaches. Although the data examined in this study are cross-sectional in nature, earlier longitudinal evaluations are informative as their results speak to the importance of age in shaping temperament presentations, and vice versa–temperament features as predictors of infant age. It should be noted that no study to date has explored the latter, that is, used temperament dimensions to classify infants with respect to their age, likely due to sample size limitations and only recently available methodological advances in empirically based classification techniques.

## Gender differences in temperament

Although a number of gender differences in temperament have been reported for older children and adults, fewer exist for children younger than one year of age [8,25,27,28]. Differences in infancy have been limited to activity level and fear/behavioral inhibition. Higher activity level and approach is evident in boys [29,30], with girls exhibiting greater hesitation in approaching novel objects [14,31]. Campbell and Eaton [29] applied meta-analytic procedures to summarize 46 studies addressing activity level in infancy, estimating the size of the gender difference at 0.2 standard deviations based on objective measures (parent-report measures estimated the difference to be smaller). Gender differences in approach-withdrawal have been reported for samples from different countries [30,32–34], with parents rating boys higher in approach. Martin et al. [31] reported a large and significant gender difference for distress to novelty, with 6-month-old girls receiving higher scores.

Gender differences also have been documented with the IBQ-R, as boys received higher scores on Activity and High Intensity Pleasure, and girls higher scores on Fear [24,25,35,36]. Infant gender also predicted intercept values of Fear trajectories, with girls demonstrating higher levels at 4 and 6 months [5,26]. Girls also started out at lower values (i.e., intercept estimates) for Activity Level, Approach, and High Intensity Pleasure. Similar to age/developmental differences research, gender-related temperament studies have only compared temperament for boys and girls, not considering gender classification based on temperament features. Importantly, age- and gender-based temperament distinctions have not been considered jointly, discerning whether age-related changes inform gender differences.

## Present study

In this study, we leveraged IBQ-R data collected across multiple laboratories ($N$ = 4,438) to further investigate age and gender differences in infancy, addressing yet unanswered questions. Specifically, algorithmic modeling techniques were used to discern the extent to which the 14 IBQ-R subscale scores (referred to as features) accurately classified participating children as boys ($n$ = 2,298) or girls ($n$ = 2,093; 47 children were missing gender data) and into three age groups: youngest ($<$ 24 weeks; $n$ = 1,102), mid-range (24 to 48 weeks; $n$ = 2,557), and oldest ($>$ 48 weeks; $n$ = 779), because of previously noted gender-based variability [14,29–34] and significant developmental differences among these age groups (e.g., with respect to brain growth and maturation; [37,38]). This study addresses an important gap in research, being the first to consider temperament attributes as determinants of age and gender groupings, quantifying the extent to which early reactivity and regulation provide the features necessary for accurate prediction. Importantly, this work also allows for simultaneous classification of age and gender categories, providing an opportunity to consider the extent to which gender differences are informed by infant age, and to our knowledge, this is the first to study to do so. That is, despite prior demonstrations of reliable age and gender differences in temperament, the two classifications have not been considered jointly, examining whether gender differences were age dependent in a single investigation. Moreover, this effort provides a new direction for infancy and temperament research, serving as a methodological demonstration of machine learning applications, not yet utilized in these areas of scientific inquiry. This meta-analytic data driven effort is the first to rely on advanced machine learning techniques using temperament features to classify infants into age and gender groups, rather than compare temperament of children who vary in age and gender, considering these classifications simultaneously. This cross-laboratory effort also overcomes prior limitations associated with small samples that were not representative, producing results circumscribed in terms generalizability.

## Materials and methods

### Measures

The Infant Behavior Questionnaire-Revised (IBQ-R; [24]. This parent-report measure of temperament was developed for infants between 3- and 12-months of age. The IBQ-R contains 191 items, which yield 14 scales: Activity Level, Smiling/Laughter, Approach, High Intensity Pleasure, Perceptual Sensitivity, Vocal Reactivity (loading onto Positive Affectivity/Surgency); Fear, Distress to Limitations, Sadness, Falling Reactivity (Negative Emotionality); Duration of Orienting, Soothability, Cuddliness/Affiliation, Low Intensity Pleasure (Regulatory Capacity/Orienting). Individual items are rated on a 7-point scale reflecting the frequency of occurrence of the behavior in the past week (two weeks for less frequent events, such as encounters with unfamiliar settings/adults). Reliability of the IBQ-R has been supported for mothers and fathers, as well as samples from different cultures, with Cronbach's α values ranging from .77

to .96 [39–41]. Evidence supports the predictive and construct validity of IBQ-R scores [42–44]. Cronbach's α values for the 14 subscales included in the current analysis, derived from 29 datasets, ranged from .74 to .89 (mean α = .82). These temperament features were used to classify children into gender and age categories via Machine Learning algorithms.

## Procedure

Data sets (*N* = 29) were acquired by emailing researchers who requested the IBQ-R or published research using the instrument between 2006 and 2019. All of the researchers had received approval from their respective Human Research Protection Programs (HRPPs)/Institutional Review Boards (IRBs) prior to initiating data collection: Human Studies Committee at the Brigham and Women's Hospital in Boston, MA and the Icahn School of Medicine at Mount Sinai in New York; IRB at Boston Children's Hospital; Pennsylvania State University IRB; Rutgers-Newark IRB; Virginia Tech IRB; University of North Carolina at Greensboro IRB; Emory IRB; University of Washington IRB Committee D; Northern Illinois University IRB #1; Brown University HRPP/IRB; IRB of the University of Minnesota's Human Research Protection Program; University of Michigan Health Sciences and Behavioral Sciences IRB; Health Sciences IRB at University of Wisconsin; Harvard University Committee on the Use of Human Subjects in Research; University of California, Irvine HRPP/IRB; University of Missouri IRB; IRB of Western Kentucky University; University of North Carolina at Chapel Hill IRB; Western Washington University IRB; University of Virginia IRB for the Social and Behavioral Sciences; Wayne State University IRB; Colorado Multiple IRB; obtaining written informed consent. Contributors were asked to provide item level data from the IBQ-R as well as infant age, gender, and race. For all participants, the IBQ-R was completed by the infant's mother. See Table 1 for a brief description of the samples.

## Analytic strategy

Descriptive statistics across gender and age groups were computed first (Table 2). We then constructed a model framework allowing us to assess the utility of fine-grained temperament dimensions with respect to gender and age classifications. This framework resulted in a total of five (5) model types, which included: 1) gender: boys vs. girls; 2) age groups: youngest (< 24 weeks) vs. mid-range (24 to 48 weeks) vs. oldest (> 48 weeks) infants; and gender by age group analyses: 3) boys vs. girls in the youngest age group; 4) boys vs. girls in the mid-range age group; 5) boys vs. girls in the oldest age group. Classification of infant gender within age groups allows us to determine if predictive strength of gender-based classification is more accurate for younger vs. older infants.

Established machine learning techniques, methodologically rigorous and shown to provide reliable/reproducible results, were used in this study (e.g., [45,46]). Specifically, for all models, we used repeated 10-fold cross-validation partitioning with random assignment: a training dataset including 70% of the sample, and 30% reserved as a hold-out dataset (testing) to evaluate the predictive utility of the trained models. A total of 11 different algorithms were considered for each model type, including: (1) linear discriminant analysis; (2) generalized linear modeling; (3) support vector machines; (4) K-nearest neighbor; (5) naïve bayes; (6) classification and regression trees; (7) C5.0 classification; (8) bootstrapped aggregated trees; (9) ensembled decision trees (Random Forest; [47,48]); (10) gradient boosting; and (11) multiclass adaptive boosting (AdaBoost). These algorithms were chosen based on their applicability and widespread use in the classification modeling literature [45,46], and in order to achieve most robust and replicable results discernable across multiple modeling techniques. The aforementioned models were then compared to discern the most effective classification of infant

**Table 1. Sample descriptions.**

| Researcher(s) | Sample Size (*N*) | Infant Age (Weeks) | Gender (% Male) | Race (% Non-White) | Sample Description |
|---|---|---|---|---|---|
| Bosquet & Wright | 668 | 20.23–63.25 | 53.3 | 71.1 | Community sample of infants |
| Gartstein | Study 1: 387<br>Study 2: 143<br>Study 3: 84<br>Study 4: 67 | 15.00–52.00<br>11.00–51.00<br>11.00–54.00<br>24.00–48.00 | 50.1<br>49.7<br>94.0<br>44.8 | NA<br>11.2<br>13.1<br>10.4 | Community sample of infants |
| Perez-Edgar, Buss, & LoBue | Study 1: 138<br>Study 2: 267 | 16.00–47.20<br>12.00–68.18 | 55.0<br>46.8 | 26.8<br>41.6 | Community sample of infants |
| Bell & Calkins | 353 | 20.57–57.00 | 49.3 | 23.8 | Community sample of healthy infants |
| Goodman | Study 1: 82<br>Study 2: 252 | 12.00–52.00<br>12.00–52.00 | 62.2<br>44.8 | 43.9<br>27.0 | Community sample of mothers with history of major depression<br>Mothers received treatment for major depression during pregnancy |
| Spieker | 221 | 22.00–40.00 | 54.8 | 81.4 | Mothers received mental health treatment during pregnancy |
| Bridgett | 178 | 16.00–48.00 | 47.2 | 29.2 | Full term, healthy infants |
| Salisbury | 172 | 23.00–32.00 | 51.7 | 47.7 | Prenatal exposure to depression, antidepressants |
| Mliner & Gunnar | 158 | 48.53–89.20 | 50.6 | 60.8 | Full term, healthy infants |
| Muzik | 157 | 23.27–44.40 | 52.2 | 43.3 | Mothers oversampled for trauma |
| Stifter | 149 | 24.57–57.29 | 53.0 | 8.1 | Community sample of full-term infants |
| Planalp | 148 | 23.00–87.00 | 48.0 | 24.3 | Community sample of infants |
| Mehr & Spelke | 123 | 11.71–88.43 | 59.3 | 32.5 | Community sample of full-term infants |
| Lukowski | 108 | 39.71–46.14 | 53.7 | 38.0 | Full term, healthy infants |
| Groh | 91 | 25.81–42.93 | 52.2 | 21.1 | Full term, healthy infants |
| Lickenbrock | 80 | 12.00–35.00 | 60.0 | 15.0 | Low-risk community sample of infants |
| Santelli | 73 | 47.57–70.14 | 47.9 | 32.9 | Vaginally delivered infants exclusively breastfed until 1 month of age |
| Du Rocher Shudlich | 73 | 24.80–58.80 | 52.1 | 16.4 | Parents living together since birth of child |
| Anzman-Frasca | 59 | 51.00–57.00 | 54.2 | 11.9 | Full term, healthy infants (a portion of the entire sample was included in this study) |
| Thrasher | Study 1: 12<br>Study 2: 28<br>Study 3: 20 | 6.33–8.67<br>6.33–9.10<br>6.80–9.00 | 73.0<br>38.0<br>45.0 | NA<br>NA<br>NA | Full term, healthy infants |
| Diaz | 47 | 40.00 | 44.7 | 23.4 | Full term, healthy infants |
| Dayton | 47 | 16.00–31.00 | 42.9 | 35.7 | High risk sample of families (e.g., poverty, violence exposure, psychopathology) |
| Moding | 43 | 26.00–102.00 | 41.9 | 34.9 | No food allergies, feeding difficulties |
| Jordan | 42 | 20.00–45.00 | 31.0 | 19.0 | Full term, healthy infants |

**Table 2. Descriptive statistics for the temperament subscales by gender and age group.**

| Models | Gender | | | | | | Age Group | | | | | | | | |
|---|---|---|---|---|---|---|---|---|---|---|---|---|---|---|---|
| | Girls | | | Boys | | | Youngest < 24 weeks | | | Mid-Range 24 to 48 weeks | | | Oldest > 48 weeks | | |
| | Mean | SD | Range | Mean | SD | Range | Mean | SD | Range | Mean | SD | Range | Mean | SD | Range |
| Activity | 4.25 | 1.11 | 0.33–6.93 | 4.29 | 1.08 | 0.47–6.80 | 4.12 | 0.89 | 0.53–6.67 | 4.29 | 1.21 | 0.33–6.93 | 4.43 | 0.87 | 0.47–6.87 |
| Approach | 4.79 | 1.39 | 0.17–7.00 | 4.84 | 1.37 | 0.17–7.00 | 3.98 | 1.50 | 0.17–7.00 | 5.00 | 1.23 | 0.33–7.00 | 5.55 | 0.91 | 1.42–7.00 |
| Smiling/ Laughter | 4.61 | 1.36 | 0.10–7.00 | 4.63 | 1.34 | 0.10–7.00 | 4.37 | 1.15 | 0.20–7.00 | 4.63 | 1.49 | 0.10–7.00 | 5.01 | 0.91 | 0.70–7.00 |
| High Intensity Pleasure | 5.32 | 1.41 | 0.09–7.00 | 5.49 | 1.38 | 0.09–7.00 | 4.98 | 1.23 | 0.55–7.00 | 5.47 | 1.53 | 0.09–7.00 | 5.95 | 0.74 | 0.27–7.00 |
| Perceptual Sensitivity | 3.27 | 1.35 | 0.08–6.83 | 3.33 | 1.36 | 0.17–7.00 | 2.89 | 1.25 | 0.17–7.00 | 3.38 | 1.40 | 0.08–7.00 | 3.71 | 1.18 | 0.42–6.83 |
| Vocal Reactivity | 4.42 | 1.38 | 0.08–7.00 | 4.41 | 1.35 | 0.17–7.00 | 3.92 | 1.10 | 0.33–7.00 | 4.43 | 1.47 | 0.08–7.00 | 5.22 | 0.89 | 1.00–7.00 |
| Distress to Limitations | 3.46 | 0.90 | 0.69–6.31 | 3.56 | 0.92 | 0.19–6.38 | 3.27 | 0.83 | 0.19–6.25 | 3.55 | 0.91 | 0.56–6.31 | 3.71 | 0.95 | 0.25–6.38 |
| Fear | 2.51 | 1.07 | 0.19–6.44 | 2.28 | 0.95 | 0.06–6.69 | 2.05 | 0.90 | 0.31–6.25 | 2.43 | 1.02 | 0.19–6.44 | 2.74 | 1.02 | 0.06–6.69 |
| Falling Reactivity | 4.57 | 1.20 | 0.23–6.92 | 4.50 | 1.19 | 0.08–7.00 | 4.63 | 1.07 | 1.08–7.00 | 4.62 | 1.03 | 1.15–7.00 | 4.13 | 1.67 | 0.08–6.92 |
| Sadness | 2.97 | 0.98 | 0.14–6.29 | 3.03 | 0.96 | 0.14–5.79 | 2.91 | 0.99 | 0.36–6.29 | 3.01 | 0.98 | 0.14–6.21 | 3.10 | 0.89 | 0.14–5.79 |
| Cuddliness | 5.12 | 1.11 | 0.53–7.00 | 5.08 | 1.13 | 0.29–7.00 | 5.39 | 1.13 | 0.76–7.00 | 5.03 | 1.12 | 0.29–7.00 | 4.87 | 0.97 | 0.41–6.82 |
| Duration of Orienting | 3.69 | 1.16 | 0.17–7.00 | 3.69 | 1.13 | 0.25–7.00 | 3.62 | 1.19 | 0.08–7.00 | 3.73 | 1.16 | 0.17–7.00 | 3.63 | 1.01 | 0.92–6.83 |
| Low Intensity Pleasure | 4.79 | 1.07 | 0.69–7.00 | 4.72 | 1.06 | 1.23–7.00 | 4.74 | 1.12 | 0.69–7.00 | 4.82 | 1.05 | 1.23–7.00 | 4.52 | 0.98 | 1.77–7.00 |
| Soothability | 4.64 | 1.07 | 0.50–7.00 | 4.58 | 1.12 | 0.39–7.00 | 5.39 | 1.13 | 0.76–7.00 | 4.62 | 1.13 | 0.50–7.00 | 4.66 | 1.09 | 0.94–7.00 |

gender and age with temperament features based on misclassification rates, Cohen's kappa coefficients, and sensitivity and specificity via the area under the curve (AUC) from Receiver Operator Curves (ROC), considered as indicators of predictive accuracy.

Misclassification provides a simplistic posterior assessment of model classification based on contingency tables and is often used for initial classification and model accuracy evaluation. Accuracy indicators, reported herein, represent the inverse of misclassification rates. Cohen's kappa coefficient assesses reliability of categorization, which incorporates chance agreement, is normalized, and can range from -1 to 1. Kappa values will typically be lower than overall misclassification indictors, as it represents a more conservative estimate given its assessment of accuracy compared to random assignment. The area under an ROC curve (area under the curve, or AUC) is a third metric used to evaluate the accuracy of binary classifiers, which encapsulates both Type I and Type II errors [49]. However, ROC-AUC is limited insofar as it does not take predicted probability values and goodness of fit of evaluated models into account. While all three indicators provide unique assessments of classification accuracy, overall misclassification rate (or, inversely, accuracy) is the most broadly used metric for classification evaluation [50]. For all of the model classification indices, higher values (i.e., closer to 1) can be considered superior, indicative of more optimal performance.

## Results

Overall, classification accuracy was superior for age relative to gender categories, based on misclassification rates (i.e., accuracy indicators), Kappa, and area under the curve (AUC) indicators (Table 3A).

Specifically, across all algorithmic models, age-based classification outperformed gender-based classification for all classification outcomes.

Gender classification was performed within the three infant age groups next (Table 3B), with classification effectiveness for gender generally superior in the oldest age group (> 48 weeks). That is, oldest age group classification models consistently outperformed others based on the AUC, and this was the case for the majority of classification algorithms with respect to accuracy and Kappa indicators. Next, we focused on the AUC, especially informative in

**Table 3A. Classification effectiveness indicators across machine learning algorithms: Gender and age-based classification with temperament features.**

| Models | Gender Classification: boys vs. girls | | | Age Classification: youngest (age < 24 weeks) vs. mid-range (age 24 to 48 weeks) vs. oldest (age > 48 weeks) | | |
|---|---|---|---|---|---|---|
| | Accuracy | Kappa | AUC | Accuracy | Kappa | AUC* |
| Linear Discriminant Analysis | .558 | .162 | .422 | .641 | .284 | .517 |
| Generalized Linear Modeling | .569 | .153 | .485 | .630 | .295 | .526 |
| Support Vector Machines | .559 | .169 | .432 | .637 | .308 | .517 |
| K-Nearest Neighbor | .556 | .084 | .471 | .650 | .271 | .529 |
| Naïve Bayes | .577 | .094 | .451 | .634 | .272 | .512 |
| Classification and Regression Trees | .565 | .099 | .424 | .645 | .240 | .514 |
| C5.0 Classification | .575 | .099 | .422 | .625 | .272 | .538 |
| Bootstrapped Aggregated Trees | .580 | .099 | .422 | .640 | .274 | .535 |
| Ensembled Decision Trees (Random Forest) | .580 | .133 | .485 | .641 | .289 | .535 |
| Gradient Boosting | .556 | .157 | .432 | .631 | .306 | .522 |
| Multi-class Adaptive Boosting (AdaBoost) | .558 | .141 | .471 | .641 | .241 | .517 |

*AUC for Age Classification analysis represents a multiclass ROC indicator, based on 3 groups.

**Table 3B. Classification effectiveness indicators across machine learning algorithms: Gender by age with temperament features.**

| Models | Age Group 1 (< 24 weeks; n = 1,102) | | | Age Group 2 (24 to 48 weeks; n = 2,557) | | | Age Group 3 (> 48 weeks; n = 779) | | |
|---|---|---|---|---|---|---|---|---|---|
| | Accuracy | Kappa | AUC | Accuracy | Kappa | AUC | Accuracy | Kappa | AUC |
| Linear Discriminant Analysis | .563 | .164 | .404 | .557 | .148 | .429 | .527 | .152 | .452 |
| Generalized Linear Modeling | .549 | .154 | .407 | .551 | .147 | .436 | .574 | .112 | .501 |
| Support Vector Machines | .530 | .185 | .439 | .559 | .130 | .463 | .608 | .093 | .525 |
| K-Nearest Neighbor | .569 | .066 | .427 | .558 | .098 | .450 | .589 | .138 | .570 |
| Naïve Bayes | .594 | .117 | .455 | .556 | .087 | .436 | .572 | .194 | .542 |
| Classification and Regression Trees | .536 | .075 | .437 | .548 | .075 | .471 | .546 | .133 | .536 |
| C5.0 Classification | .567 | .087 | .457 | .573 | .112 | .436 | .571 | .159 | .487 |
| Bootstrapped Aggregated Trees | .572 | .092 | .410 | .568 | .060 | .422 | .618 | .093 | .565 |
| Ensembled Decision Trees (Random Forest) | .577 | .105 | .386 | .559 | .109 | .451 | .584 | .138 | .552 |
| Gradient Boosting Method | .540 | .123 | .395 | .567 | .155 | .405 | .540 | .214 | .576 |
| Multi-class Adaptive Boosting (AdaBoost) | .563 | .119 | .404 | .557 | .131 | .429 | .527 | .100 | .452 |

capturing differences for gender classification models across age groups because of its long-standing widespread use for comparative purposes in the machine learning classification literature [51] and visualization capabilities (Figs 1–3). AUC gender classification indicators were superior for the oldest age group, yielding higher values across different algorithmic models, illustrated in Fig 3.

## Discussion

We set out to leverage existing IBQ-R datasets from multiple laboratories (N = 4,438) to address an important gap in research by investigating age and gender classifications in early childhood, and overcoming limitations of the published studies such as small sample sizes that cannot be considered representative or provide widely generalizable results. Relying on algorithmic modeling techniques, 14 IBQ-R subscale scores served as features used to classify participating children as boys (n = 2,298) and girls (n = 2,093), and into three age groups:

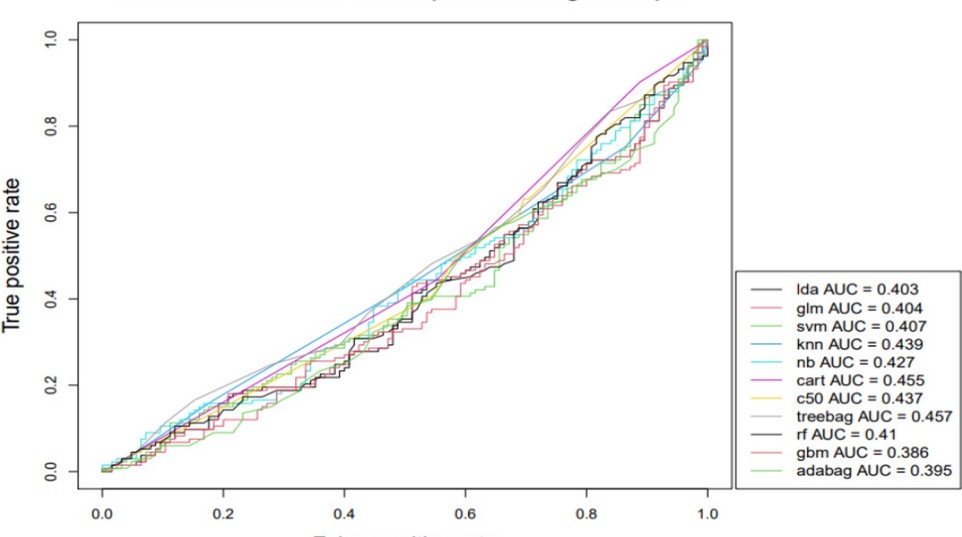

**Fig 1.** <u>Note</u>: lda—Linear Discriminant Analysis; glm—Generalized Linear Modeling; svm—Support Vector Machines; knn—K-Nearest Neighbor; nb—Naïve Bayes; cart—Classification and Regression Trees; c50—C5.0 Classification; treebag—Bootstrapped Aggregated Trees; rf—Ensembled Decision Trees (Random Forest); gbm—Gradient Boosting Method; adabag—Multi-class Adaptive Boosting (AdaBoost).

youngest (< 24 weeks; n = 1,102), mid-range (24 to 48 weeks; n = 2,557), and oldest (> 48 weeks; n = 779). Importantly, this approach allowed us to simultaneously classify infants into age and gender categories, providing an opportunity for the first time to consider the extent to which gender differences are informed by infant age. This study also makes an important contribution to the literature as a novel methodological demonstration. That is, the present

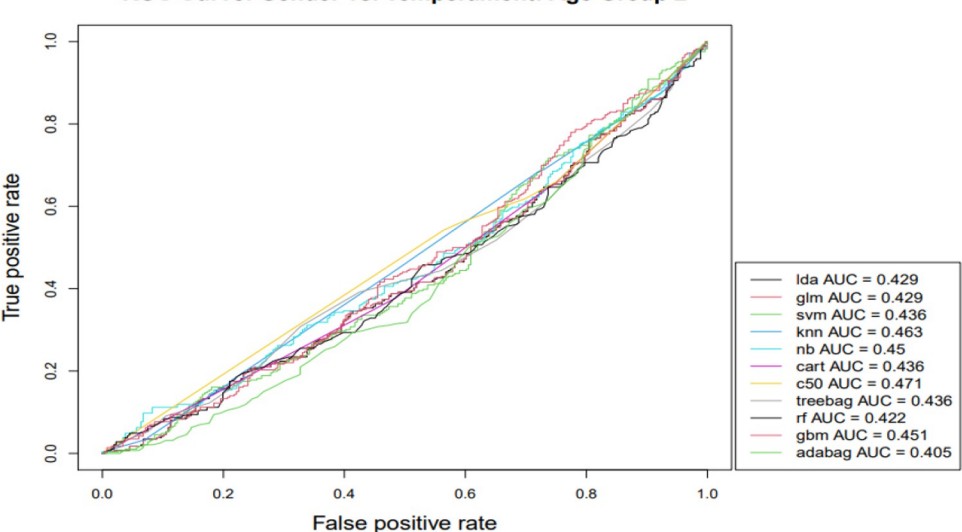

**Fig 2.** <u>Note</u>: lda—Linear Discriminant Analysis; glm—Generalized Linear Modeling; svm—Support Vector Machines; knn—K-Nearest Neighbor; nb—Naïve Bayes; cart—Classification and Regression Trees; c50—C5.0 Classification; treebag—Bootstrapped Aggregated Trees; rf—Ensembled Decision Trees (Random Forest); gbm—Gradient Boosting Method; adabag—Multi-class Adaptive Boosting (AdaBoost).

**ROC Curve: Gender vs. Temperament: Age Group 3**

lda AUC = 0.45
glm AUC = 0.452
svm AUC = 0.501
knn AUC = 0.525
nb AUC = 0.57
cart AUC = 0.542
c50 AUC = 0.536
treebag AUC = 0.487
rf AUC = 0.565
gbm AUC = 0.552
adabag AUC = 0.576

**Fig 3.** <u>Note</u>: lda—Linear Discriminant Analysis; glm—Generalized Linear Modeling; svm—Support Vector Machines; knn—K-Nearest Neighbor; nb—Naïve Bayes; cart—Classification and Regression Trees; c50—C5.0 Classification; treebag—Bootstrapped Aggregated Trees; rf—Ensembled Decision Trees (Random Forest); gbm—Gradient Boosting Method; adabag—Multi-class Adaptive Boosting (AdaBoost).

application of machine learning algorithms provides a new direction for infancy and temperament research, as well as meta-analytic investigations more broadly.

Results based on accuracy indicators (the inverse of misclassification rates), Cohen's kappa coefficients, and AUC (incorporating sensitivity and specificity parameters) demonstrated that temperament features provided superior classification of age groups relative to gender, which is consistent with the existing literature insofar as age effects have generally been more robust (e.g., not dependent on methodology; [5,26,52]). As noted, gender differences in infancy have been largely limited to activity level and fear/behavioral inhibition, with higher activity level and approach reported for boys [29,30] and greater fear/behavioral inhibition for girls [14,25,31,35,36]. These gender differences are somewhat controversial due to a lack of consensus regarding their origin (i.e., biologically based or largely a function of socialization; [53]) and questions regarding the role of parental expectations. That is, parents could rate boys and girls differently not due to actual variability in behavior but as a function of their own culturally influenced ideas about what is typical behavior in boys vs. girls. This explanation cannot be ruled out completely, although existing research suggests that gender differences are not entirely dependent on methodology (i.e., have been identified via behavioral observations along with parent report; [33,52]).

Importantly, gender classification by age groups results suggest this is most effective for the oldest age group, in line with the literature that indicates gender differences in temperament attributes become more pronounced with age [54]. Although a number of factors could be contributing to this pattern of results—accentuated gender differences in temperament with increasing age, and, conversely more accurate classification of gender with temperament features for oldest participants—socialization is often described as critical among these. The primary mechanism invoked in such explanations involves the infants' interactional history, and is consistent with literature that indicates mothers respond differently to their sons and daughters [55–59], presenting with different affordances as social interaction partners (e.g., [60]). Over time, such differences could result in divergent trajectories with respect to temperament

due to differences in socialization goals/approaches for boys vs. girls. Specifically, parents may prioritize relationship orientation for daughters, but competence and autonomy for sons [61–63]. These and other socialization-related pathways may be responsible for the stronger temperament-based classification of boys and girls later in infancy observed herein.

At the same time, gender is viewed as a marker for a host of sex-linked distinctions in physiological processes. For example, prenatal exposure to high levels of androgen is predictive of later behavior problems, primarily of the externalizing type (e.g., ADHD; [64]), and used to explain early vulnerability observed in boys with respect to this set of problems [65]. Postpartum biological effects are also possible, for example via testosterone increases for boys in infancy, referred to as "mini-puberty," peaking by the second month and returning to baseline at about 6 months [66]. Sex-linked differentiation in brain structures and functions occurs with maturation, resulting in greater discrepancies with age. For example, Goldstein et al. [67] reported that the amygdala tends to be larger in males and the hippocampus larger in females (see Hines [68] for a related review).

Follow-up analyses outlining feature importance for classification models were performed for the Ensembled Decision Trees (Random Forest) to further interpretation of the observed results. Random Forest methods provide an effective mechanism for feature selection and importance using tree-based mechanisms to rank node classification via the mean decrease in gini impurity, i.e., the probability that a random sample in a particular tree node would be mislabeled using the distribution of the node sample, averaged across all trees [69]. Figures provided in Supplemental Materials (S1–S3 Figs) demonstrate that while Fear was the most important feature in distinguishing boys and girls for the youngest and mid-range age group, for oldest infants, low intensity pleasure was most influential. In fact, for youngest infants (S3 Fig), all three distress-related scales (Fear, Distress to Limitations, Sadness) were of primary importance in classifying infants accurately by gender via the Random Forest algorithm. Positive emotionality and regulatory dimensions of temperament (e.g., Falling Reactivity, Approach) begin to take on greater importance for mid-range and oldest infants. Notably, certain temperament features detracted from model accuracy in classifying infants by gender (i.e., associated with lowest negative importance values), particularly Cuddliness, Vocal Reactivity, and Smiling and Laughter in the youngest age group and Smiling and Laughter, Perceptual Sensitivity, and Activity in the oldest age group. These results identify the temperament attributes that did not differentiate boys and girls effectively, and it is of interest that the list of these poorly differentiating features varied by age. When the most important features were considered for age classification and gender classification models only, Fear again emerged as the critical dimension, which is in line with the extensive literature documenting the developmental progression as well as gender differences for this domain of temperament [2,13,14,26,54].

This work is not without limitations, chief among these our reliance on a single method (i.e., parent report) in the assessment of infant temperament. Future studies should aggregate datasets providing different sources of information, including behavioral observations and physiological measures, such as cortisol reactivity, heart rate variability/respiratory sinus arrhythmia, and/or frontal alpha asymmetry ascertained via electroencephalogram (EEG) recordings. In addition, the outcomes examined in this study were limited to child gender and age. Future studies with older children should conduct classification analyses with additional dependent variables, particularly symptom and disorder classifications (e.g., clinical/subclinical/asymptomatic ADHD). It should be noted that we did not consider classification based on race/ethnicity because of a far more limited literature suggesting these differences can be discerned on the basis of temperament, and future research should examine related models, as relevant studies accumulate. Finally, the present modeling approach could be extended and

potentially improved by applying ensembling modeling approaches (i.e., using multiple algorithms simultaneously), as opposed to relying on singular modeling frameworks.

This study underscores the importance of meta-analytic investigations and cross-laboratory collaborations, providing illusive answers to questions, such as those related to intersections of gender and age in temperament development, that have not been previously addressed. Because of the large cross-laboratory sample included herein, this study provides most optimal comparative data for the IBQ-R (Table 2), which has emerged as a widely used infant temperament assessment tool. Importantly, the present investigation serves as a methodological illustration for application of machine learning techniques in infancy and temperament research, as well as developmental science more broadly. Given the propensity for differing algorithmic methods to have strengths and weaknesses that may bias predictive outcomes and classification accuracy, we selected 11 established algorithmic modeling and classification techniques to quantify the most robust outcomes, simultaneously demonstrating the viability of machine learning approaches in this area of scientific inquiry. Results of this study make an important contribution to developmental temperament research, demonstrating effective age group classification on the basis of fine-grained temperament features, and indicating more effective gender classification for the older age group, with multiple implications for future mechanistic research examining potential socialization and biological contributors.

## Supporting information

**S1 Fig. Note.** DL–distress to limitations; Sad–sadness; PS–perceptual sensitivity; App–approach; Fall–falling reactivity; DO–duration of orienting; HP–high intensity pleasure; LP–low intensity pleasure; Act–activity level; Sooth–soothability; SL–smiling and laughter; VR–vocal reactivity; Cud–cuddliness.
(TIF)

**S2 Fig. Note.** Fall–falling reactivity; HP–high intensity pleasure; LP–low intensity pleasure; Sad–sadness; VR–vocal reactivity; App–approach; DL–distress to limitations; SL–smiling and laughter; PS–perceptual sensitivity; DO–duration of orienting; Sooth–soothability; Cud–cuddliness; Act–activity level.
(TIF)

**S3 Fig. Note.** LP–low intensity pleasure; App–approach; VR–vocal reactivity; Fall–falling reactivity; Sad–sadness; DL–distress to limitations; Cud–cuddliness; DO–duration of orienting; Sooth–soothability; HP–high intensity pleasure; Act–activity level; PS–perceptual sensitivity; SL–smiling and laughter.
(TIF)

## Author Contributions

**Conceptualization:** Maria A. Gartstein, D. Erich Seamon, Jennifer A. Mattera, Samuel A. Mehr.

**Data curation:** Maria A. Gartstein, Jennifer A. Mattera, Michelle Bosquet Enlow, Rosalind J. Wright, Koraly Perez-Edgar, Kristin A. Buss, Vanessa LoBue, Martha Ann Bell, Sherryl H. Goodman, Susan Spieker, David J. Bridgett, Amy L. Salisbury, Megan R. Gunnar, Shanna B. Mliner, Maria Muzik, Cynthia A. Stifter, Elizabeth M. Planalp, Samuel A. Mehr, Elizabeth S. Spelke, Angela F. Lukowski, Ashley M. Groh, Diane M. Lickenbrock, Rebecca Santelli, Tina Du Rocher Schudlich, Stephanie Anzman-Frasca, Catherine Thrasher, Anjolii Diaz, Carolyn Dayton, Kameron J. Moding, Evan M. Jordan.

**Formal analysis:** D. Erich Seamon.

**Funding acquisition:** Michelle Bosquet Enlow, Rosalind J. Wright, Koraly Perez-Edgar, Kristin A. Buss, Vanessa LoBue, Martha Ann Bell, Sherryl H. Goodman, Susan Spieker, David J. Bridgett, Amy L. Salisbury, Megan R. Gunnar, Maria Muzik, Cynthia A. Stifter, Elizabeth M. Planalp, Samuel A. Mehr, Elizabeth S. Spelke, Angela F. Lukowski, Ashley M. Groh, Diane M. Lickenbrock, Rebecca Santelli, Tina Du Rocher Schudlich, Stephanie Anzman-Frasca, Catherine Thrasher, Anjolii Diaz, Carolyn Dayton, Kameron J. Moding, Evan M. Jordan.

**Investigation:** Maria A. Gartstein, Michelle Bosquet Enlow, Rosalind J. Wright, Koraly Perez-Edgar, Kristin A. Buss, Vanessa LoBue, Martha Ann Bell, Sherryl H. Goodman, Susan Spieker, David J. Bridgett, Amy L. Salisbury, Megan R. Gunnar, Shanna B. Mliner, Maria Muzik, Cynthia A. Stifter, Elizabeth M. Planalp, Samuel A. Mehr, Elizabeth S. Spelke, Angela F. Lukowski, Ashley M. Groh, Diane M. Lickenbrock, Rebecca Santelli, Tina Du Rocher Schudlich, Stephanie Anzman-Frasca, Catherine Thrasher, Anjolii Diaz, Carolyn Dayton, Kameron J. Moding, Evan M. Jordan.

**Methodology:** Maria A. Gartstein, D. Erich Seamon, Jennifer A. Mattera, Michelle Bosquet Enlow, Rosalind J. Wright, Koraly Perez-Edgar, Kristin A. Buss, Vanessa LoBue, Martha Ann Bell, Sherryl H. Goodman, Susan Spieker, David J. Bridgett, Amy L. Salisbury, Megan R. Gunnar, Maria Muzik, Cynthia A. Stifter, Elizabeth M. Planalp, Samuel A. Mehr, Elizabeth S. Spelke, Angela F. Lukowski, Ashley M. Groh, Diane M. Lickenbrock, Rebecca Santelli, Tina Du Rocher Schudlich, Stephanie Anzman-Frasca, Catherine Thrasher, Anjolii Diaz, Carolyn Dayton, Kameron J. Moding, Evan M. Jordan.

**Project administration:** Maria A. Gartstein, Jennifer A. Mattera, Michelle Bosquet Enlow, Rosalind J. Wright, Koraly Perez-Edgar, Kristin A. Buss, Vanessa LoBue, Martha Ann Bell, Sherryl H. Goodman, Susan Spieker, David J. Bridgett, Amy L. Salisbury, Megan R. Gunnar, Shanna B. Mliner, Maria Muzik, Cynthia A. Stifter, Elizabeth M. Planalp, Samuel A. Mehr, Elizabeth S. Spelke, Angela F. Lukowski, Ashley M. Groh, Diane M. Lickenbrock, Rebecca Santelli, Tina Du Rocher Schudlich, Stephanie Anzman-Frasca, Catherine Thrasher, Anjolii Diaz, Carolyn Dayton, Kameron J. Moding, Evan M. Jordan.

**Resources:** Michelle Bosquet Enlow, Rosalind J. Wright, Koraly Perez-Edgar, Martha Ann Bell, Sherryl H. Goodman, Susan Spieker, David J. Bridgett, Amy L. Salisbury, Megan R. Gunnar, Maria Muzik, Elizabeth M. Planalp, Samuel A. Mehr, Elizabeth S. Spelke, Angela F. Lukowski, Ashley M. Groh, Diane M. Lickenbrock, Rebecca Santelli, Tina Du Rocher Schudlich, Stephanie Anzman-Frasca, Catherine Thrasher, Anjolii Diaz, Carolyn Dayton, Kameron J. Moding, Evan M. Jordan.

**Software:** D. Erich Seamon.

**Visualization:** D. Erich Seamon.

**Writing – original draft:** Maria A. Gartstein, D. Erich Seamon, Jennifer A. Mattera, Michelle Bosquet Enlow, Rosalind J. Wright, Koraly Perez-Edgar, Kristin A. Buss, Vanessa LoBue, Martha Ann Bell, Sherryl H. Goodman, Susan Spieker, David J. Bridgett, Amy L. Salisbury, Megan R. Gunnar, Shanna B. Mliner, Maria Muzik, Cynthia A. Stifter, Elizabeth M. Planalp, Samuel A. Mehr, Elizabeth S. Spelke, Angela F. Lukowski, Ashley M. Groh, Diane M. Lickenbrock, Rebecca Santelli, Tina Du Rocher Schudlich, Stephanie Anzman-Frasca, Catherine Thrasher, Anjolii Diaz, Carolyn Dayton, Kameron J. Moding, Evan M. Jordan.

**Writing – review & editing:** Maria A. Gartstein, D. Erich Seamon, Jennifer A. Mattera, Michelle Bosquet Enlow, Rosalind J. Wright, Koraly Perez-Edgar, Kristin A. Buss, Vanessa LoBue, Martha Ann Bell, Sherryl H. Goodman, Susan Spieker, David J. Bridgett, Amy L. Salisbury, Megan R. Gunnar, Shanna B. Mliner, Maria Muzik, Cynthia A. Stifter, Elizabeth M. Planalp, Samuel A. Mehr, Elizabeth S. Spelke, Angela F. Lukowski, Diane M. Lickenbrock, Rebecca Santelli, Tina Du Rocher Schudlich, Stephanie Anzman-Frasca, Catherine Thrasher, Anjolii Diaz, Carolyn Dayton, Kameron J. Moding, Evan M. Jordan.

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
