## [Decision Letter · Decision Letter 0]

4 Nov 2021

PONE-D-21-33028Using Machine Learning to Understand Age and Gender Classification Based on Infant TemperamentPLOS ONE

Dear Dr. Gartstein,

Thank you for submitting your manuscript to PLOS ONE. After careful consideration, we feel that it has merit but does not fully meet PLOS ONE’s publication criteria as it currently stands. Therefore, we invite you to submit a revised version of the manuscript that addresses the points raised during the review process.

The idea of the paper is good and has potential to create new knowledge in that research area.Authors should provide more clearer explanation about  the motivation of the study, research problem and contribution of this research.Two reviewers provided their comments to improve the quality of the paper.

We look forward to receiving your revised manuscript.

Kind regards,

Siuly Siuly, PhD

Academic Editor

PLOS ONE

Journal Requirements:

Reviewers' comments:

Reviewer's Responses to Questions

**Comments to the Author**

1. Is the manuscript technically sound, and do the data support the conclusions?

Reviewer #1: Yes

Reviewer #2: Yes

2. Has the statistical analysis been performed appropriately and rigorously? 

Reviewer #1: Yes

Reviewer #2: Yes

3. Have the authors made all data underlying the findings in their manuscript fully available?

Reviewer #1: No

Reviewer #2: Yes

4. Is the manuscript presented in an intelligible fashion and written in standard English?

Reviewer #1: Yes

Reviewer #2: Yes

5. Review Comments to the Author

Reviewer #1: By applying machine learning techniques, this paper analyzed the temperament features of infants and classified these features into gender and age categories. It is a significant improvement of traditional analytical investigation and a very meaningful research area for young children under 12 months, yet a few comments.

(1) In the section "Analytic Strategy" line 254, it mentioned the "gender by age group analyses". However, there is no Table or Figure in the paper to further elucidate these model types. Please try to provide supplemental information or move this group to the "Discussion" section as future work.

(2) For Table 3a, "Gender and age-based classification with temperament features", three metrics, i.g. Accuracy, Kappa, AUC, are applied to evaluate each of 11 machine learning algorithms. I hesitate to ask, what features do you use to do Gender Classification and Age Classification? I thought the features were those 14 models listed in Table 2, line 294. If not, please further explain. If yes, how to integrate 14 features' results into one metric?

Reviewer #2: This manuscript represents a meta-analysis utilizing IBQ-R data collected across multiple laboratories to overcome the limitations of smaller samples in elucidating links among temperament, age, and gender in early childhood. This paper is generally well written, logical and discusses a hot topic. The results also present a good effectiveness. I think this paper can be accepted.

6. PLOS authors have the option to publish the peer review history of their article (what does this mean?). If published, this will include your full peer review and any attached files.

Reviewer #1: No

Reviewer #2: No

---

## [Author Response · Author response to Decision Letter 0]

28 Dec 2021

November 14, 2021

Dr. Siuly, 

Academic Editor

PLOS ONE

Dear Dr. Siuly,

Thank you for the opportunity to revise and resubmit our manuscript, entitled: “Using Machine Learning to Understand Age and Gender Classification Based on Infant Temperament.” We appreciate your appraisal of the idea behind our paper as good, with potential to create new knowledge in the research area. In this revision, we provide a clearer explanation regarding the motivation behind the study, the research problem addressed, and the contribution of this research in the Introduction and Discussion sections (pgs. 10, 22, 26, 27). With respect to data sharing, there are ethical/legal restrictions on sharing the de-identified data set used in this study imposed by contributors’ institutions (e.g., Institutional Review Board, Human Research Protection Program, Office of Research). In fact, the first author and her institution had to enter into Memorandum of Understanding agreements with a number of contributing sites in order to obtain the relevant data. These types of arrangements can be considered upon request to access data to the first and corresponding author: Maria A. Gartstein (gartstma@wsu.edu). I would also like to note that the ethical guidelines of the American Psychological Association were closely followed in conducting research presented herein. We have included the full names of the ethics committees that approved data collection across the 29 sites, indicating that all co-authors/investigators obtained written informed consent.

Reviewers’ comments and recommendations are addressed in detail below.

Reviewer #1: By applying machine learning techniques, this paper analyzed the temperament features of infants and classified these features into gender and age categories. It is a significant improvement of traditional analytical investigation and a very meaningful research area for young children under 12 months, yet a few comments.

(1) In the section "Analytic Strategy" line 254, it mentioned the "gender by age group analyses". However, there is no Table or Figure in the paper to further elucidate these model types. Please try to provide supplemental information or move this group to the "Discussion" section as future work.

####################################################################

We apologize if this element of the results was confusing, as Table 3b in fact presents gender by age group findings, one component of which (AUC indicators across considered algorithmic models) is also illustrated in Figure 1a-1c. We have clarified the latter for the reader in the revision, as requested (pgs. 19-20). ####################################################################

(2) For Table 3a, "Gender and age-based classification with temperament features", three metrics, e.g. Accuracy, Kappa, AUC, are applied to evaluate each of 11 machine learning algorithms. I hesitate to ask, what features do you use to do Gender Classification and Age Classification? I thought the features were those 14 models listed in Table 2, line 294. If not, please further explain. If yes, how to integrate 14 features' results into one metric?

####################################################################

The reviewer is correct insofar as the features are the 14 temperament scales, namely: Activity Level, Smiling/Laughter, Approach, High Intensity Pleasure, Perceptual Sensitivity, Vocal Reactivity, Fear, Distress to Limitations, Sadness, Falling Reactivity, Duration of Orienting, Soothability, Cuddliness/Affiliation, and Low Intensity Pleasure. On the other hand, Accuracy, Kappa, AUC, are indicators used to evaluate the predictive accuracy of the 11 machine learning algorithms considered in this study that rely on the 14 temperament features for gender, age, and gender by age classifications. In the revision, we have clarified related language (pgs. 10, 16, 22), increasing clarity according to this recommendation. ####################################################################

Reviewer #2: This manuscript represents a meta-analysis utilizing IBQ-R data collected across multiple laboratories to overcome the limitations of smaller samples in elucidating links among temperament, age, and gender in early childhood. This paper is generally well written, logical and discusses a hot topic. The results also present a good effectiveness. I think this paper can be accepted.

####################################################################

We thank the reviewer for this positive view of our manuscript.

####################################################################

We hope that you and the reviewers find this manuscript worthy of publication in PLOS ONE.

---

## [Decision Letter · Decision Letter 1]

14 Mar 2022

Using Machine Learning to Understand Age and Gender Classification Based on Infant Temperament

PONE-D-21-33028R1

Dear Dr. Gartstein,

We’re pleased to inform you that your manuscript has been judged scientifically suitable for publication and will be formally accepted for publication once it meets all outstanding technical requirements.

Kind regards,

Siuly Siuly, PhD

Academic Editor

PLOS ONE

Additional Editor Comments (optional):

Reviewers' comments:

Reviewer's Responses to Questions

**Comments to the Author**

1. If the authors have adequately addressed your comments raised in a previous round of review and you feel that this manuscript is now acceptable for publication, you may indicate that here to bypass the “Comments to the Author” section, enter your conflict of interest statement in the “Confidential to Editor” section, and submit your "Accept" recommendation.

Reviewer #1: All comments have been addressed

Reviewer #2: All comments have been addressed

2. Is the manuscript technically sound, and do the data support the conclusions?

Reviewer #1: Yes

Reviewer #2: Yes

3. Has the statistical analysis been performed appropriately and rigorously? 

Reviewer #1: Yes

Reviewer #2: Yes

4. Have the authors made all data underlying the findings in their manuscript fully available?

Reviewer #1: Yes

Reviewer #2: Yes

5. Is the manuscript presented in an intelligible fashion and written in standard English?

Reviewer #1: Yes

Reviewer #2: Yes

6. Review Comments to the Author

Reviewer #1: In this revised version, major changes are made in Introduction, Materials and Methods, and Discussion.

(a) In the Introduction session, the disadvantage of existing classification method is elaborated.

(b) In the Materials and Methods session, detailed researcher list is given.

(c) In the Discussion session, contributions of this paper is emphasized and the ensembling modeling approach is mentioned as the potential improvement in further work.

Other typos are clear in this version. Good job.

Reviewer #2: This manuscript represents a meta-analysis utilizing IBQ-R data collected across multiple laboratories to overcome the limitations of smaller samples in elucidating links among temperament, age, and gender in early childhood. This paper is generally well written, logical and discusses a hot topic. The results also present a good effectiveness. I think this paper can be accepted.

7. PLOS authors have the option to publish the peer review history of their article (what does this mean?). If published, this will include your full peer review and any attached files.

Reviewer #1: No

Reviewer #2: No

---

## [Editor Report · Acceptance letter]

21 Mar 2022

PONE-D-21-33028R1 

Using Machine Learning to Understand Age and Gender Classification Based on Infant Temperament 

Dear Dr. Gartstein:

I'm pleased to inform you that your manuscript has been deemed suitable for publication in PLOS ONE. Congratulations! Your manuscript is now with our production department. 

Kind regards, 

on behalf of

Dr. Siuly Siuly 

Academic Editor

PLOS ONE